# Nuclear translocation of mitochondrial dehydrogenases as an adaptive cardioprotective mechanism

Shubhi Srivastava[1], Priyanka Gajwani [1], Jordan Jousma [2], Hiroe Miyamoto[2], Youjeong Kwon[2], Arundhati Jana[3], Peter T. Toth [2,4], Gege Yan[2,3], Sang-Ging Ong [2,3] ✉ & Jalees Rehman [1,2,3,5] ✉

Chemotherapy-induced cardiac damage remains a leading cause of death amongst cancer survivors. Anthracycline-induced cardiotoxicity is mediated by severe mitochondrial injury, but little is known about the mechanisms by which cardiomyocytes adaptively respond to the injury. We observed the translocation of selected mitochondrial tricarboxylic acid (TCA) cycle dehydrogenases to the nucleus as an adaptive stress response to anthracycline-cardiotoxicity in human induced pluripotent stem cell-derived cardiomyocytes and in vivo. The expression of nuclear-targeted mitochondrial dehydrogenases shifts the nuclear metabolic milieu to maintain their function both in vitro and in vivo. This protective effect is mediated by two parallel pathways: metabolite-induced chromatin accessibility and AMP-kinase (AMPK) signaling. The extent of chemotherapy-induced cardiac damage thus reflects a balance between mitochondrial injury and the protective response initiated by the nuclear pool of mitochondrial dehydrogenases. Our study identifies nuclear translocation of mitochondrial dehydrogenases as an endogenous adaptive mechanism that can be leveraged to attenuate cardiomyocyte injury.

Anthracyclines such as doxorubicin remain a central part of treatment regimens in certain cancers with high mortality, such as triple-negative breast cancer[1,2]. However, anthracyclines have off-target effects and can cause cardiovascular damage, thus increasing iatrogenic morbidity and mortality as well as limiting their use in patients at risk for developing cardiomyopathy[3–5]. These incidences can even exceed those reported for recurrence of the underlying cancer[4,6,7].

Much of the current research on anthracycline-cardiotoxicity has focused on the mechanisms of mitochondrial injury, which involve mitochondrial DNA damage, mitochondrial oxidative stress, anthracycline binding to cardiolipins of the mitochondrial inner membrane, the opening of the mitochondrial transition pore, and apoptosis[8]. Despite extensive efforts, drugs to mitigate mitochondrial damage in cardiomyopathies, such as mitochondrial antioxidants, are still not clinically approved[9,10]. A complementary approach could be to identify endogenous adaptive mechanisms in response to anthracycline-induced mitochondrial injury and leverage such endogenous protective responses, but little is known about such putative adaptive responses. The nuclear genome encodes the vast majority of mitochondrial proteins; therefore, it is likely that adaptive responses to mitochondrial injury involve the communication of mitochondrial injury to the nucleus to restore mitochondrial health[11–16]. Mitochondrial TCA cycle enzymes can be found in the nucleus, and their levels increase during stress[17,18]. Therefore, the nuclear presence of mitochondrial TCA cycle enzymes may represent a possible mode of mitochondria-to-nucleus signaling that could mediate adaptive

[1]Department of Biochemistry and Molecular Genetics, The University of Illinois College of Medicine, Chicago, IL, USA. [2]Department of Pharmacology and Regenerative Medicine, The University of Illinois College of Medicine, Chicago, IL, USA. [3]Division of Cardiology, Department of Medicine, The University of Illinois College of Medicine, Chicago, IL, USA. [4]Research Resources Center, University of Illinois, Chicago, IL, USA. [5]University of Illinois Cancer Center, Chicago, IL, USA. ✉e-mail: sangging@uic.edu; jalees@uic.edu

regulatory responses, as metabolites generated by mitochondrial enzymes can serve as cofactors for epigenetic regulation[19,20].

Here we show that in response to anthracycline-induced toxicity, cardiomyocytes initiate an endogenous protective response via selective translocation of mitochondrial TCA cycle dehydrogenases into the nucleus. This adaptive response protects the cells from anthracycline-induced cell death and maintains the function of surviving cardiomyocytes in vitro and in vivo.

## Results

### Translocation of TCA cycle enzymes to the nucleus

To investigate the endogenous protective mechanisms in anthracycline-induced cardiotoxicity, we first differentiated cardiomyocytes from two distinct human iPSC lines, as this is a widely accepted model for chemotherapy-induced cardiotoxicity[21–24], and increased their purity using glucose deprivation, which eliminates non-cardiomyocytes[21]. The hiPSC-derived cardiomyocytes (hiPSC-CMs) were characterized by immunofluorescence staining for actinin and troponin-T (Supplementary Fig. 1a). We hypothesized that anthracycline-induced cell death likely begins when the toxicity exhausts endogenous protective mechanisms. Exposure of hiPSC-CMs using varying durations and doses of doxorubicin (DRN) was performed to identify the induction of cell stress and cell death. We found that 1 μM DRN induced cell death at 24 h but less cell death at 6 h (Supplementary Fig. 1b). Interestingly, DRN-induced DNA damage was present as early as 6 h (Supplementary Fig. 1c), which suggested a temporal window from 6 to 24 h during which the endogenous protective response may avert cell death.

We posited that mitochondrial distress may be communicated to the nucleus via changes in the nuclear metabolic compartment, as mitochondrial enzymes are known to translocate to the nucleus, and the metabolites they generate can modulate gene expression by acting as epigenetic cofactors[25,26]. We analyzed the localization of the following mitochondrial TCA cycle enzymes: pyruvate dehydrogenase subunit 1 (PDH-E1), malate dehydrogenase 2 (MDH-2), isocitrate dehydrogenase 2 (IDH-2), and citrate synthase (CS), all of which generate epigenetic cofactor metabolites[25,26]. We observed that following DRN exposure, the mitochondrial TCA cycle enzymes PDH-E1, MDH-2, and IDH-2 increasingly translocated to the nucleus from 6 to 24 h, whereas CS was retained in mitochondria (Fig. 1a, b and Supplementary Fig. 2a, c). The specific translocation of selected dehydrogenases (PDH-E1, MDH-2, and IDH-2) but not of CS further suggests that the translocation of mitochondrial enzymes was not the result of non-specific mitochondrial barrier disruption. CoxIV and Troponin-T (TnT) were stained as mitochondrial and cardiomyocyte markers, respectively. The absence of CS, CoxIV, and TnT in the nuclei again highlights the intactness of the organelle barriers and compartments during this early stage of the stress response and further underscores the remarkable specificity of selective nuclear translocation of PDH-E1, MDH-2, and IDH-2. Quantification indicated that ~10% of these mitochondrial enzymes translocated into the nucleus following DRN-induced stress, with no change in the subcellular localization of CS (Supplementary Fig. 2b, d–f). To rule out two-dimensional imaging artifacts, we performed three-dimensional z-stack reconstruction of confocal images and observed the presence of TCA cycle dehydrogenases in the nucleus following DRN stress, whereas the mitochondrial inner membrane CoxIV, and TnT remained outside the nucleus (Fig.1c–f and Supplementary Fig. 2g). Interestingly, irrespective of whether the mitochondrial network was perinuclear (Fig.1d) or at the cell periphery (Fig.1f), the TCA cycle dehydrogenases accumulated in the nucleus in response to DRN-stress, whereas in control cells, MDH-2 (Fig.1c) and IDH-2 (Fig.1e) were exclusively found in the mitochondria. This highlighted that the localization of TCA cycle dehydrogenases in the nucleus occurred only under stress and was not due to visual artifacts caused by perinuclear mitochondria, which are present even under control conditions. As shown in Supplementary Movies 1–4, the 3D reconstruction confirmed that the enzyme signals and DAPI signals (Alexa 594 pseudo-colored in green and DAPI in blue) were imaged from the same focal plane, showing their co-localization upon doxorubicin stress.

We further validated the results using sucrose-gradient centrifugation subcellular fractionation, where the enzymes were detected in the pure isolated nuclear fractions after DRN treatment and were absent in the nuclei isolated from control cells (Fig. 1g). Troponin T and Kir2.1 were used as cytoplasmic markers. No detection of CoxIV (mitochondrial inner membrane), CS (mitochondrial matrix), TnT, or Kir2.1 validated the purity of the nuclear fractions. As hiPSC-CMs differentiated for relatively short periods of time and may be only partially mature[27] and exhibit altered sensitivity toward stressors[28], we also maintained hiPSC-CMs for 60 days to enhance maturation and observed similar translocation of MDH-2 after DRN stress, whereas CS remained confined to the mitochondria (Supplementary Fig. 3a–d).

To investigate whether the observed nuclear translocation of mitochondrial enzymes in response to DRN also occurred in vivo, we used a murine model of DRN-induced acute cardiotoxicity. C57/BL6 mice were injected with a total of 20 mg/kg body weight DRN intraperitoneally, which induces acute cardiotoxicity at 92 h[29]. We surmised that adaptive enzyme translocation would precede cardiotoxicity and thus obtained myocardial sections at 72 h post-DRN injection. As in hiPSC-CMs, IDH-2 was redistributed to the nucleus of cardiomyocytes following in vivo anthracycline treatment in cardiomyocytes isolated from mice treated with DRN (Fig.1h, i, Supplementary Fig. 3e, f, and Supplementary Movies 5, 6), and also in cardiac tissue sections (Supplementary Fig. 4a–d and Supplementary Movies 7, 8). To investigate whether the observed effects were conserved across multiple anthracyclines, we also treated hiPSC-CMs with the anthracyclines daunorubicin or idarubicin for 24 h. As with doxorubicin, we detected TCA cycle enzyme translocation (MDH-2) to the nuclei of hiPSC-CMs treated with daunorubicin and idarubicin (Supplementary Fig. 5).

We next assessed whether the mechanism of translocation involved the co-translational transport of mitochondrial proteins and thus blocked protein translation by cycloheximide prior to DRN treatment. If the nuclear presence of mitochondrial enzymes was due to de novo protein synthesis instead of translocation of previously synthesized enzymes from the mitochondria to the nucleus, nuclear detection of TCA cycle dehydrogenases post-DRN exposure would be abolished. As shown in Supplementary Fig. 6a, b, we observed translocation of TCA PDH-E1 post-DRN treatment even in the presence of cycloheximide using a dose which effectively blocked de novo protein synthesis (Supplementary Fig. 6c).

### Mitochondrial dehydrogenases are protective in the nucleus

To investigate whether the nuclear translocation of TCA cycle enzymes serves as an adaptive cardioprotective response or potentially mediates the detrimental effects of anthracyclines, we designed a lentiviral tetracycline-inducible vector in which the mitochondria targeting signal (MTS) of selected TCA cycle dehydrogenases was replaced by a nuclear localization signal (NLS) in MDH-2, IDH-2 and three subunits of the PDH complex (PDHC). An HA tag was added at the C-terminus of MDH-2 and IDH-2 to differentiate the nuclear pool of the enzyme from the mitochondrial pool (Supplementary Fig. 7a). Pyruvate dehydrogenase (PDH subunit E1) functions in a complex with two other enzymes (dihydrolipoamide acyltransferase subunit E2 and dihydrolipoyl dehydrogenase subunit E3). To express all three subunits in the nucleus and assemble a nuclear PDH complex, we designed a construct where E1, E2, and E3 were linked together by two self-cleavable linkers, T2A and P2A[30,31]. The co-translational self-cleavage of these linkers leads to the independent folding of the three subunits, and the NLS assists in the translocation of the three subunits into the nucleus, thus creating a nuclear PDH complex. The three subunits E1,

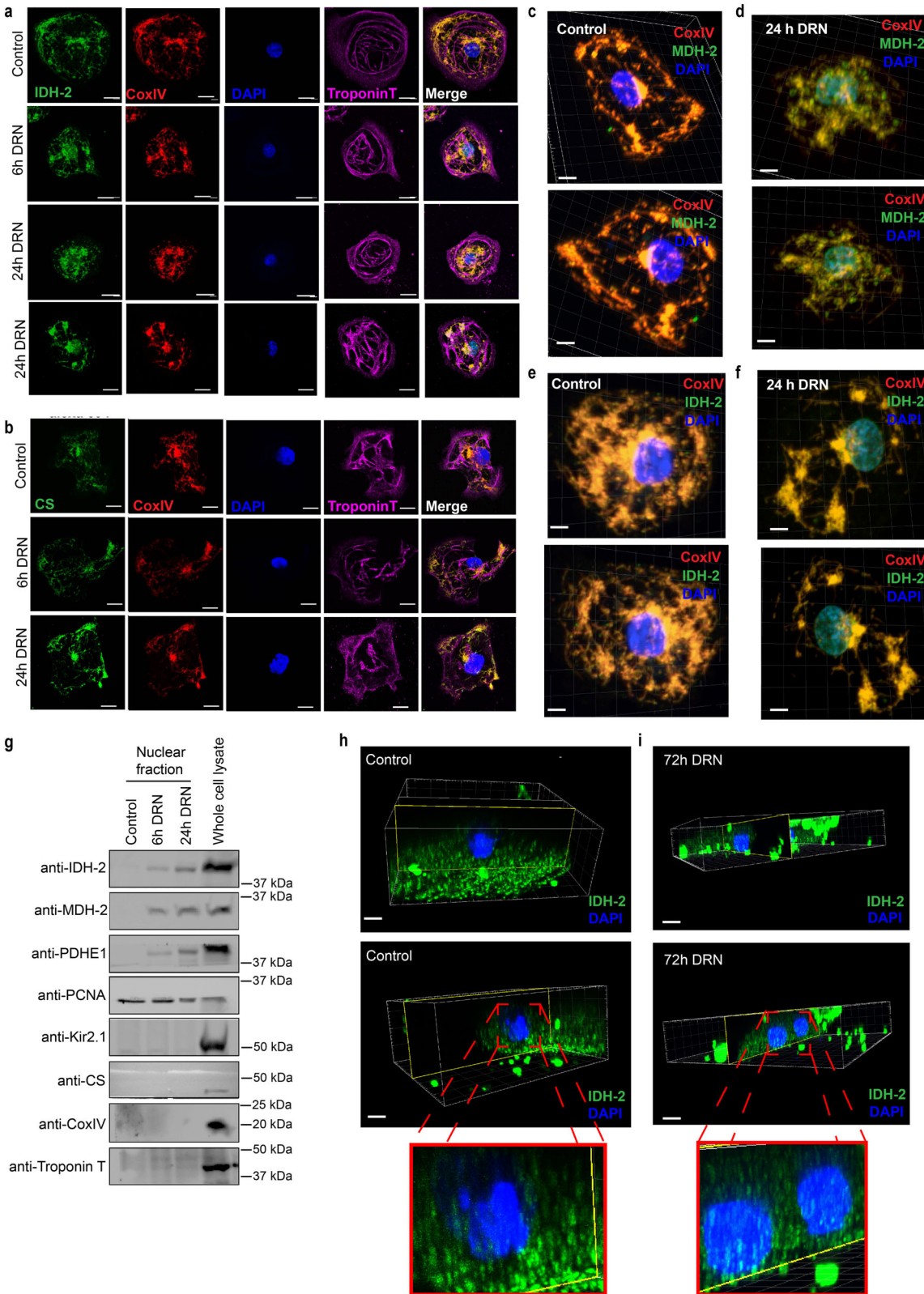

E2, and E3 were tagged by Flag, HA, and V5 tags at the C-terminus, respectively, to differentiate them from the mitochondrial pool (Supplementary Fig. 7a). In this system, the TCA cycle dehydrogenases are expressed in the nucleus without affecting the mitochondrial pool of the enzymes under the Tet$_{ON}$ state (Supplementary Fig. 7b–g).

To assess whether nuclear-targeted IDH-2, MDH-2, and PDHC remained functional in the nucleus, we isolated nuclei from cells transduced with these NLS constructs. The enzymatic activity was measured as the conversion of NAD$^+$ to NADH for MDH-2 and PDHC and the conversion of NADP$^+$ to NADPH for IDH-2 (Supplementary Fig. 8a–c). As shown in Fig. 2a–c, we observed that these enzymes were active in the nucleus. The activity of PDHC depends on the close proximity and activity of all three subunits, which suggested that the observed conversion of NAD$^+$ to NADH was due to the NLS PDHC

**Fig. 1 | Mitochondrial TCA cycle enzymes translocate to the nucleus upon doxorubicin treatment. a, b** Confocal microscopy images of hiPSC-derived cardiomyocytes treated with 1 μM doxorubicin for 6 or 24 h (DRN) or DMSO (control). The cells were stained with anti-isocitrate dehydrogenase (IDH-2) (**a**) or anti-citrate synthase (CS) (**b**) and detected with a secondary antibody conjugated with Alexa 594. Cells were stained with CoxIV and Troponin-T (TnT) as markers for mitochondria and cardiomyocytes, respectively. The signals from CoxIV and TnT were detected with secondary antibodies conjugated with Alexa-633 and Alexa-488, respectively. Scale bar: 20 μm. **c–f** 3-dimensional reconstituted z-stack images of control (**c, e**) and doxorubicin-treated cells (**d, f**) stained for MDH-2 (**c, d**) and IDH-2 (**e, f**), CoxIV, and TnT. Scale bar: 10 μm. **g** Immunoblots demonstrating the nuclear localization of IDH-3, PDH-E1, and MDH-2 in the nuclear fraction from cells treated with doxorubicin (DRN) for 6 or 24 h or from DMSO-treated control cells. TnT and Kir2.1 were probed to rule out the contamination of cytoplasmic fractions, and CoxIV was used as a mitochondrial marker in the isolated nuclear lysate. Whole-cell lysates from untreated cells were used as another control. **h, i** Immunostained images of cardiomyocytes isolated from mice treated with vehicle (**h**) or doxorubicin (**i**) for 72 h, cells stained for IDH-2, CoxIV, and TnT. Scale bar: 10 μm; images are represented along with the three-dimensional reconstruction of z-stack images.

reconstituting the whole complex within the nucleus of transduced cells. The nuclear intervention of mitochondrial dehydrogenases does not affect the mitochondrial pool of enzymes, ensuring that the phenotype observed after induced nuclear targeting was attributed to the nuclear protein levels, not by the mitochondrial pool. We also measured the NAD+/NADH or NADP$^+$/NADPH ratio in isolated nuclei. The reduced nuclear NAD+/NADH ratio or NADP$^+$/NADPH ratio further confirms that the tetracycline-inducible nuclear targeting system preserves the catalytic activity of the TCA cycle enzymes (Supplementary Fig. 8d–f). The reduction in nuclear pyruvate levels and an increase in nuclear acetyl CoA levels (Supplementary Fig. 8g, h), further confirmed the functional activity of the NLS-PDH complex in the nucleus.

To investigate how nuclear TCA cycle enzymes impact DRN-induced cell death, we transduced hiPSC-CM with NLS constructs and induced nuclear translocation 24 h prior to DRN stress. After 24 h of tetracycline-induction of the NLS constructs (and corresponding tetracycline exposure in control cells and DRN-treated cells to control for the direct effect of tetracycline), cells were exposed to DRN for 24 h and stained with Annexin V to detect apoptosis. As shown in Fig. 2d, DRN exposure increased apoptotic cell death of cardiomyocytes, which was markedly attenuated by nuclear targeting of IDH-2. We observed that nuclear expression of IDH-2, PDHC, and MDH-2 significantly reduced the release of the cardiomyocyte injury marker Troponin I[32] (Fig. 2e), whereas a control nuclear-targeted NLS-GFP-2A-NLS-mCherry construct under a tetracycline-inducible promoter (Supplementary Fig. 8i) showed no cardioprotection (Fig. 2f). As NLS-IDH-2 showed the most robust protective response against DRN-toxicity, we focused on NLS-IDH-2 for subsequent experiments. We measured the protection against DNA damage in cells expressing nuclear IDH-2 in DRN-treated cells and observed a reduction in DNA damage caused by DRN-toxicity in the presence of nuclear IDH-2 (Fig. 2g). We also analyzed the expression levels of anti-apoptotic proteins and cleaved caspase 9. As shown in Fig. 2h–k, pre-emptive nuclear targeting of IDH-2 increased the expression of anti-apoptotic mRNA and protein levels and reduced activation of mitochondrial apoptosis (Fig. 2l and Supplementary Fig. 8j–m). Together, these results indicate that the mitochondria-to-nucleus translocation of specific mitochondrial dehydrogenases serves as an endogenous protective mechanism which reduces cardiomyocyte damage.

## Nuclear IDH-2 prevents cardiac dysfunction

We next investigated the cardioprotective effects of NLS-IDH-2 on cardiomyocyte Ca$^{2+}$ handling and contractility following doxorubicin stress. As shown in Fig. 3a, b, DRN exposure reduced the amplitude of intracellular Ca$^{2+}$ influx (represented as ΔF/F$_0$), loss of rhythmic cytosolic Ca$^{2+}$ increase and decrease, and reduced overall intracellular Ca$^{2+}$, showing reduced Ca$^{2+}$ buffering compared to control cells. The nuclear expression of IDH-2 prior to DRN treatment maintained the amplitude and rhythmicity of Ca$^{2+}$ flux. Importantly, the maintenance of Ca$^{2+}$ handling following DRN stress with NLS-IDH2 expression corresponded to the preservation of cardiomyocyte contractility and rhythmicity (Fig. 3c–h).

To investigate whether nuclear expression of IDH-2 also prevented DRN-cardiotoxicity in vivo, we subcloned HA-tagged NLS-IDH-2 under α-MHC (myosin heavy chain) promoter to express IDH-2 in the nuclei of only cardiomyocytes (α-MHC-NLS-IDH-2) and not in any other organ as assessed by its expression in the lung (Supplementary Fig. 9). We analyzed left ventricular ejection fraction (LVEF) and fractional shortening (FS) as measures of cardiac function, and doxorubicin expectedly reduced LVEF and FS (Fig. 3i–k). We injected lentiviruses carrying α-MHC-NLS-IDH-2 i.v., which did not affect LVEF and FS at baseline in PBS-treated mice (Supplementary Fig. 10), however when expressed prior to doxorubicin treatment, we observed that cardiomyocyte-specific nuclear expression of IDH-2 preserved cardiac function following DRN stress in vivo (Fig.3l–n). Together, these findings suggest that nuclear IDH-2 provides protection against doxorubicin-cardiotoxicity both in vitro and in vivo.

## α-KG mediates cardioprotection by nuclear IDH-2

We next sought to determine the underlying mechanism of the protective effect of nuclear-localized IDH-2. Since we observed that the nuclear-targeted IDH-2 remains enzymatically active in the nucleus, we measured the α-KG levels in isolated nuclei extracted from cells expressing NLS-IDH-2 and indeed observed an increase in nuclear α-KG in cells expressing nuclear IDH-2 compared to uninduced cells, confirming the production of nuclear α-KG by active nuclear IDH-2 (Fig. 4a). As a signaling molecule downstream of α-KG, we focused on AMPK as a major stress-related protein that is modulated in DRN-toxicity, and on Sirt-1 as one of the major transcriptional regulators known to have an anti-apoptotic cardioprotective effect and is activated by AMPK[33–36]. We posited that α-KG generated by nuclear IDH-2 could be part of the protective mechanism, α-KG can activate AMPK, which in turn can enhance the activity of Sirt-1[37] and thus activate the AMPK-sirtuin axis, or it can also act as an epigenetic regulator by acting as a cofactor for DNA and histone methylases, thereby regulating chromatin accessibility for the active transcription of genes related to the protective effect of NLS-IDH-2 (Fig. 4b). We tested the phosphorylation of AMPK at Thr-172 as a measure of AMPK activation[38]. Interestingly, we observed an increase in phospho-AMPK in cells transduced with NLS-IDH-2 prior to DRN treatment (Fig. 4c and Supplementary Fig. 11a), and an increase in sirtuin activity in cells transduced with NLS-IDH-2 (Fig. 4d).

To further investigate the role of the AMPK-sirtuin axis as an underlying pathway in the anti-apoptotic protective response of nuclear IDH-2, we inhibited AMPK by compound C and sirtuin 1(Sirt-1) by sirtinol (Fig. 4e) and measured the activation of caspase 9 and cell death. As shown in Fig. 4f, g, nuclear expression of IDH-2 prior to DRN treatment reduced caspase 9 activity and cell death. In contrast, the inhibition of AMPK by compound C or the inhibition of Sirt-1 by sirtinol in cells expressing nuclear IDH-2 increased caspase 9 activity and cell death post-DRN treatment, whereas sirtinol or compound C alone showed no effect on cellular toxicity, suggesting that the cell death effects were not caused by the concentrations of compound C or sirtinol. To further investigate the underlying mechanism, we measured the expression of the anti-apoptotic mediator *BCL2*. We observed increased *BCL2* expression at the mRNA level in cells expressing NLS-

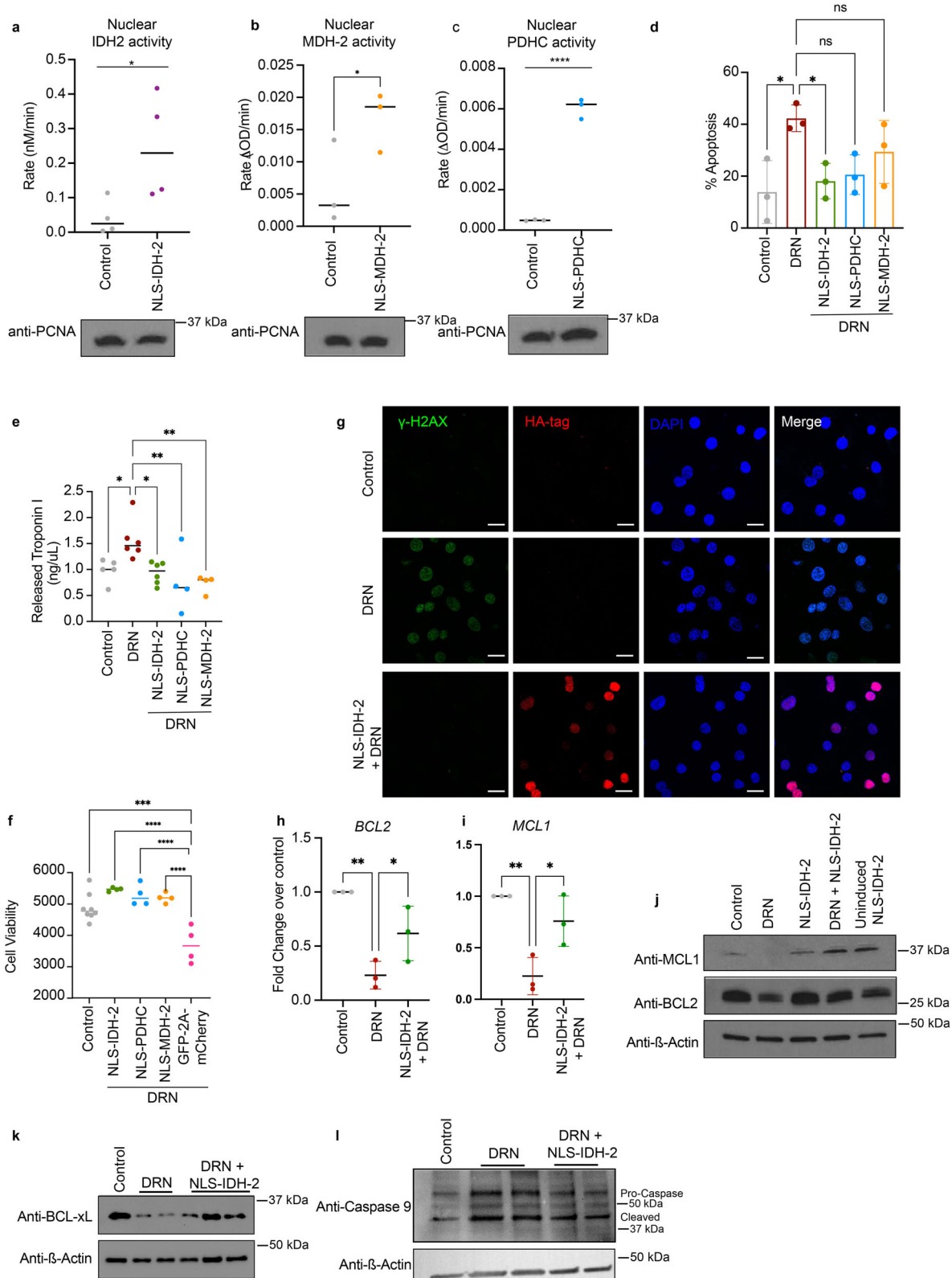

IDH-2, and *BCL2* downregulation when the cells were treated with compound C or sirtinol (Fig. 4h). We further analyzed the expression of *GLUT1* regulated by AMPK as a measure of active AMPK signaling, and we observed upregulation of these genes in NLS-IDH-2-transduced cells (Supplementary Fig. 11b).

To assess the role of α-KG in NLS-IDH2-mediated protection against DRN-toxicity, we treated cells expressing nuclear IDH-2 with

succinate, a metabolite that inhibits α-KG[39]. As shown in Fig.4i, we observed a reversal in apoptotic cell death upon succinate treatment of DRN-exposed cells expressing nuclear IDH-2. Interestingly, treatment of hiPSC-CMs with cell-permeant α-KG showed the same phenotype as nuclear expression of IDH-2, suggesting the role of α-KG as a key player in the protective response of NLS-IDH-2. We further targeted mutants of IDH-2 in the nucleus that are known to convert α-KG

**Fig. 2 | Nuclear translocation of TCA cycle enzymes prevents doxorubicin-mediated cellular damage. a–c** Enzyme activity of IDH-2 ($n = 4$) (**a**), MDH-2 ($n = 3$) (**b**), and PDHC ($n = 3$) (**c**) in nuclei isolated from cells transduced with tetracycline-inducible NLS-IDH-2 constructs. Activity is represented as the rate in nM/min or ΔOD/min, *p* value* <0.05, *p* value**** = 0.0001, and lysates were probed for PCNA to detect equal nuclear protein amounts used for the assay. **d** Reduced apoptotic cell death as measured by Annexin:V:FITC staining in cells expressing nuclear TCA cycle dehydrogenases prior to doxorubicin DRN treatment. $N = 3$ *p* value* <0.05 (**e**) Quantification of troponin I released from cardiomyocytes as a marker of cardiac injury, minimum $n = 4$ *p* value** <0.007, *p* value* <0.05. **f** Cell death as measured by Cell Titer Glo™ in cells expressing nuclear dehydrogenases prior to DRN treatment. minimum $n = 4$ *p* value*** = 0.0001, *p* value**** <0.0001. NLS-EGFP-2A-NLS-mCherry was used as a control for the 2 A linker system. **g** DNA damage as assessed by γ-H2AX foci formation. Scale bar 20 μm. **h, i** qRT–PCR showing the expression of anti-apoptotic genes *BCL2* (**h**) and *MCL1* (**i**). $n = 3$ *P* value ** <0.005, *<0.05. **j–l** Protein levels of anti-apoptotic proteins (**j, k** along with reduced caspase 9 cleavage (**l**) in cells expressing nuclear IDH-2 compared to cells treated with doxorubicin.

to D-2-hydroxyglutarate (D-2-HG)[40], thereby reducing α-KG levels (Supplementary data Fig. 11c). We observed that mutant NLS-IDH-2 (NLS-IDH-2^R140Q and NLS-IDH-2^R140QR172K) did not protect the cells from DRN-mediated apoptotic cell death, which further shows the importance of α-KG in NLS-IDH-2 mediated protection from DRN-stress (Fig. 4j).

### Nuclear IDH-2 mediated regulation of chromatin accessibility

TCA cycle metabolite signaling plays an important role in regulating gene expression through epigenetic modification of histones and DNA via acetylation or methylation[19,26,41]. α-KG acts as a cofactor for Jumonji domain-containing histone demethylases and ten-eleven translocation (TET) DNA demethylases[25]. Therefore, α-KG can also regulate chromatin accessibility to increase the expression of anti-apoptotic genes. Hence, we measured the levels of histone and DNA methylation in cells expressing nuclear IDH-2. We observed a reduction in DNA and histone methylation (Supplementary Fig. 11d, e) following the expression of nuclear IDH-2. To analyze the effect of NLS-IDH-2 on chromatin accessibility, we selected different targets based on previously published reports on genes involved in cardiac protection and performed ATAC-qPCR. These genes are related to different aspects of cellular physiology such as contractility (*TNT, KCNH2, MYOMESIN*) and calcium handling (*RYR2, CASQ, ATP2A2*), metabolism (*G6PD, HK2*), antioxidant defenses (*MNSOD, ALDR1, COX2, HO1*), and anti-apoptotic programs (*BCL2, BCLXL*) that are known to affect the cardiomyocyte health in disease conditions[42–45].

We initially tested three different time-points post-tetracycline induction to assess chromatin accessibility for 4 selected genes and observed that chromatin was accessible 18 h post-tetracycline-induction of nuclear expression of IDH-2 (Supplementary Fig. 11f–i). We then expanded the panel of analyzed chromatin regions to cover a broad range of cardiomyocyte functions and selected the 18 h time point for analyzing chromatin accessibility via ATAC-qPCR. As shown in Fig. 5, we observed more accessible chromatin corresponding to the promoter region of *HO1, BCLXL, G6PD, HK-2, KCNH2, CASQ, TNT, and BCL2* after tetracycline-induced nuclear expression of IDH-2. However, the promoter region of *MYOMESIN* was constitutively open irrespective of the nuclear presence of IDH-2. This result shows that nuclear expression of IDH-2 epigenetically regulates chromatin accessibility to affect the expression of specific cardioprotective genes.

### Nuclear IDH-2 mediated changes in chromatin accessibility correspond to gene expression changes

To evaluate whether shifts in chromatin accessibility were associated with changes in mRNA expression, we analyzed the expression levels of the genes for which we had assessed the chromatin accessibility in Fig. 5. We observed a significant change in mRNA levels 24 h post-tetracycline-induction as assessed by RNA-qPCR, but not at 18 h post-tetracycline-induction (Supplementary Fig. 11j), this suggests that the open chromatin and significant mRNA production are temporally coupled. As shown in Fig. 6, we observed increased mRNA levels of *HO1, BCLXL, G6PD, KCNH2, CASQ, TNT, MYOMESIN*, and *BCL2*. However, the mRNA levels of *COX2, MNSOD, HK-2*, and *ATP2A2* did not change, whereas *ALDR1* mRNA level were reduced upon tetracycline-induced nuclear expression of IDH-2.

Taken together, we observed four types of promoter accessibility:mRNA pairs. (1) *Accessible chromatin and altered mRNA*. We observed more accessible chromatin in promoter regions corresponding to *HO1, BCLXL, G6PD, KCNH2, CASQ, TNT*, and *BCL2* after tetracycline-induced nuclear expression of IDH-2. In line with the accessibility of chromatin, we observed a significant enrichment of mRNA for *HO1, BCLXL, G6PD, CASQ, TNT*, and *BCL2*, suggesting that these genes are actively transcribed through the open chromatin. (2) *Inaccessible chromatin and no change in mRNA*. chromatin region corresponding to promoters for *COX2, MNSOD, ALDR1*, and *ATP2A2* remains closed. In congruence with the inaccessible chromatin for *COX2, MNSOD, ALDR1*, and *ATP2A2*, mRNA levels do not change significantly upon nuclear expression of IDH-2. (3) *Constitutively accessible chromatin and altered mRNA levels*. We observed that the promoter region for *MYOMESIN* remains accessible in both uninduced and induced conditions. Even though, the promoter region for *MYOMESIN* remains constitutively active, we observed an increase in mRNA levels which shows active transcription to facilitate the maintenance of the contractility function of cardiomyocytes in cells expressing nuclear IDH-2. (4) *Accessible chromatin but no change in mRNA*. We observed open chromatin corresponding to the promoter region of *HK2*; however, the mRNA of *HK2* did not change between uninduced and induced samples, suggesting that even though the promoter region is accessible, the RNA is not actively transcribed to have a significant effect on mRNA levels (Figs. 5, 6). Together, these data show that α-KG is a metabolite that mediates nuclear IDH-2 mediated cardiac protection against doxorubicin. α-KG regulates chromatin accessibility and can also activate the AMPK-Sirt-1 signaling pathway to reduce cell death and maintain cardiac function in surviving cardiomyocytes.

Since metabolic changes in any cellular compartment can affect the whole-cell metabolite ratios, we determined the intrinsic whole-cell and nuclear levels of key metabolites that are known to have an inhibitory effect on α-KG in cells expressing nuclear IDH-2 or cells treated with DRN. We measured the whole-cell and nuclear levels of succinate, fumarate, and malate. As shown in Supplementary Fig. 12a, we observed an increase in whole-cell succinate levels in DRN-treated cells and a decrease in whole-cell succinate levels in cells expressing nuclear IDH-2 (NLS-IDH-2). We did not observe any change in nuclear succinate levels (Supplementary Fig. 12b). No significant difference was observed in whole-cell or nuclear fumarate or malate levels in cells treated with DRN; however, we observed a reduction in whole-cell malate upon expression of IDH-2 in the nucleus and a reduction in the nuclear level of fumarate (Supplementary Fig. 12c–f). These results indicate that in DRN-treated cells, the metabolites that oppose the function of α-KG, such as succinate, are high, and nuclear expression of an enzyme that can actively produce α-KG causes a shift in cellular α-KG as well as in the nucleus, where the metabolite can have an immediate effect on epigenetic remodeling of chromatin accessibility.

These results establish that nuclear IDH-2 generates nuclear α-KG, which acts as an epigenetic cofactor to increase chromatin accessibility. Increased chromatin accessibility at the promoter regions of anti-apoptotic genes such as *BCL2* can result in increased expression and help reduce cardiomyocytes. Increased accessibility of regions

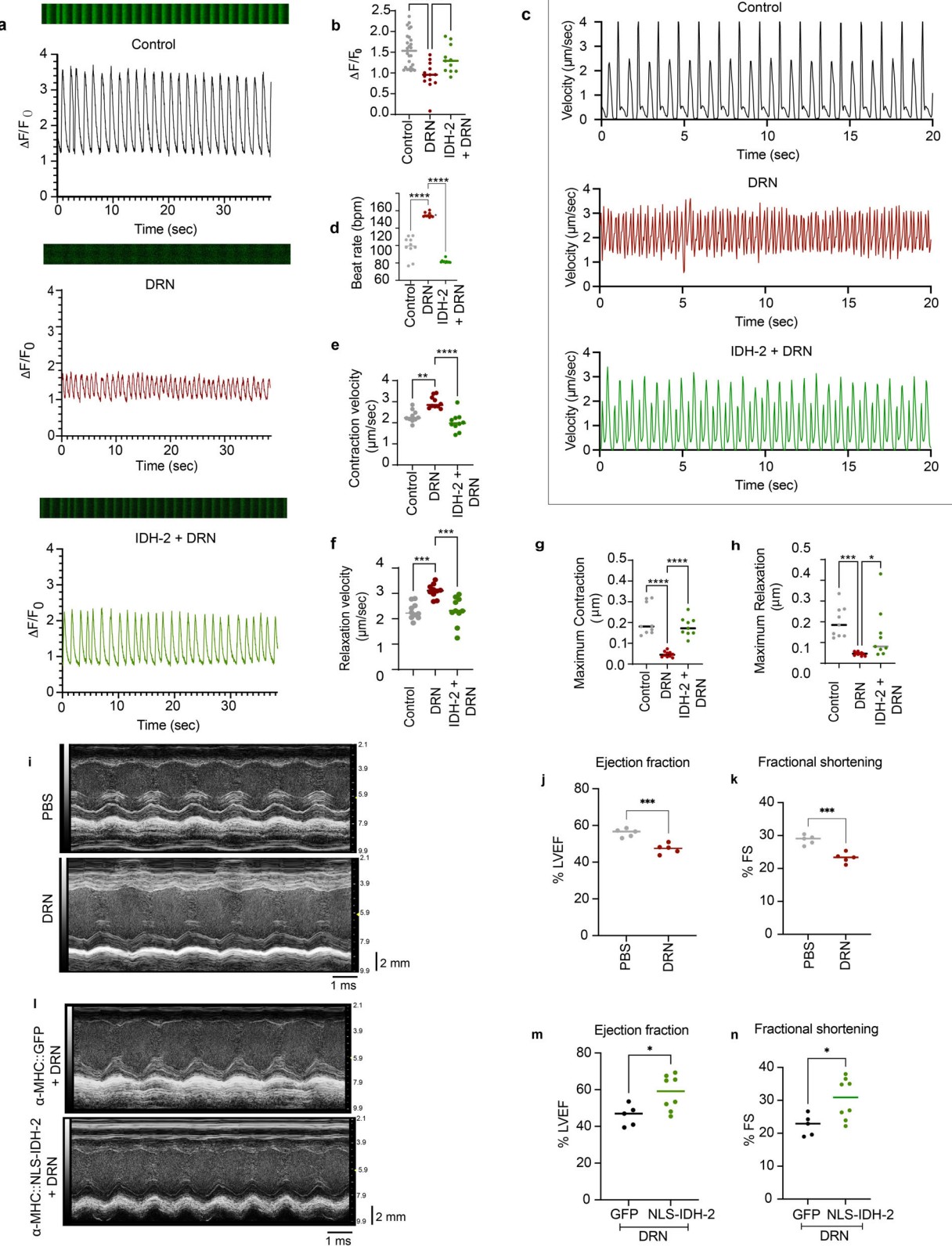

corresponding to promoters associated with contractility and calcium handling maintains the function of cardiomyocytes in DRN-induced cellular stress. In a parallel pathway, α-KG activates AMPK, which in turn activates protective sirtuins. Such adaptive mechanisms could be leveraged to shift the balance between the initial injury and adaptive responses, thus reducing the overall incidence and extent of cardiotoxicity.

## Discussion

Our study identifies mitochondria-nucleus communication as an endogenous protective stress response in anthracycline-induced cardiomyopathy both in vitro using hiPSC-CMs and in vivo using an anthracycline-cardiotoxicity mouse model. It is possible that such endogenous adaptive mechanisms are also present in other types of mitochondrial stresses and may apply to other cardiomyopathies that

**Fig. 3 | Expression of nuclear TCA dehydrogenase maintains the contractility of iPSC-CMs. a, b** Calcium handling as assessed by Fluo-4 staining in cells expressing nuclear IDH-2 in the presence of DRN. IDH-2 expression in the nucleus prior to DRN treatment maintains the rhythmic change in calcium (**a, b**). $n = 3$ $P$ value ****<0.0001, *<0.05. **c–h** Representative graph for contraction and relaxation cycle of cardiomyocyte contractility (**c**), quantification of beat rate (**d**), contraction velocity (**e**), relaxation velocity (**f**), maximum contraction (**g**), and maximum relaxation (**h**) in cells expressing nuclear IDH-2 prior to DRN treatment compared to control cells treated with DRN. $n = 3$ $P$ value ****<0.0001, ***<0.0005, **<0.005.

*=0.0399, **i** Representative M-mode images of DRN- or PBS-treated mice. **j** Left ventricle ejection fraction percentage and **k** fractional shortening recorded from M-mode tracings in mice treated with DRN or PBS. $n = 3$ $P$ value*** <0.001. **l** Representative M-mode images of DRN-treated mice injected with GFP- or NLS-IDH2-expressing lentivirus. **m** Left ventricle ejection fraction percentage and **n** fractional shortening recorded from M-mode tracings in mice expressing GFP or NLS-IDH-2 prior to DRN treatment. $P$ value *<0.05. $n = 5$ (GFP, both male and female), $n = 8$ (NLS-IDH-2, both male and female).

involve DNA damage and apoptosis. Although mitochondria-to-nucleus communication is emerging as an important pathway linking metabolism and gene regulation, relatively little is known about the mechanism of protein translocation from mitochondria to the nucleus. Possible mechanisms of translocation of TCA cycle enzymes into the nucleus upon anthracycline stress could involve the transfer of enzymes within mitochondria-derived vesicles or the direct fusion of mitochondrial membrane to the nuclear membrane, similar to contact sites that have been previously described between mitochondria and other organelles[46,47] [48–50]. Another potential application of our findings may be the explanation of why anthracycline cardiotoxicity occurs only in a subset of patients. There may be patient heterogeneity in the capacity to activate the adaptive translocation of protective mitochondrial enzymes to the nucleus. The NLS constructs we generated could aid additional studies of how TCA cycle metabolites in the nucleus induce protection not just in cardiomyocytes but also in other cell types and disease models. Our findings provide a foundation for utilizing nuclear metabolite-pool targeted approaches to reduce cellular stress. We realize that gene therapy using lentiviral constructs that we developed for our experimental studies may not be directly translatable into patient care, but safer delivery vectors, as well as small molecule-based approaches which achieve similar nuclear metabolite shifts, could be developed and increase the translational relevance of our findings.

In summary, our results unveil specific mitochondrial TCA cycle enzyme translocation as an adaptive form of mitochondria-to-nucleus communication in the setting of cardiac mitochondrial stress.

## Methods
### Animals
All in vivo experiments were performed under protocols approved by the Office of Animal Care and Institutional Biosafety (OACIB) at the University of Illinois at Chicago. C57/BL6 wild-type mice were procured from The Jackson Laboratory (JAX®) and housed in sterile static microisolator cages with sterile corncob bedding. The animals were fed ad-libitum and kept under standard light and dark cycles (14 h light and 10 h dark). For the endpoint experiments, mice were euthanized by cervical dislocation under anesthesia (i.p., injection of Ketamine 100 mg/kg body weight and Xylazine 5 mg/kg body weight).

### Human iPSC differentiation
hiPSC were generated under the protocol approved by the Institutional Review Board (IRB# #2018-0583). hiPSC-derived cardiomyocytes were differentiated from two healthy hiPSC lines using a previously described protocol[21]. Briefly, at almost 90% confluency, cells were incubated in RPMI 1640 medium (Thermo Fisher, 11875095) supplemented with B27 supplement minus insulin (Thermo Fisher, A1895601) and 6 µM CHIR99021 to initiate differentiation. On day 2 post differentiation initiation (DI), the medium was replaced with RPMI supplemented with B27 supplement minus insulin (B27-ins) without CHIR. On day 3 post-DI, the medium was replaced with RPMI + (B27-ins) with 5 µM IWR-1 (Wnt antagonist) to induce mesodermal transition. On day 5 post-DI, the medium was replaced with RPMI+ (B27-ins) with no inhibitor. On day 7 post-DI, the medium was replaced with RPMI

supplemented with B27 plus insulin (B27+ins) (Thermo Fisher, 17504044). By day 10 post-DI, we observed a rhythmic beating of differentiated cardiomyocytes. From day 10 post-DI to day 15 post-DI, the purity of cardiomyocytes was increased by selecting cardiomyocytes in RPMI + B27 medium with no glucose. The cardiomyocytes were replated in a monolayer using TrypLE Express (Life Technologies). The cells were maintained in RPMI + (B27+ins) medium for 30 days or for 60 days and dissociated using TrypLE as required for the experiments.

### Plasmids and lentivirus production
The lentiviral plasmids carrying NLS-tagged TCA cycle enzymes (pLV-TRE3G::NLS-IDH2-HA-NLS, pLV-TRE3G::NLS-PDH-E1-Flag-NLS:T2A:NLS-PDHE2-HA-NLS:P2A:NLS-PDHE3-V5-NLS, pLV-TRE3G::NLS-MDH2-HA-NLS, pLV-α-MHC::NLS-IDH2-HA-NLS, pLV-α-MHC::EGFP, pLV-TRE::NLS-IDH2$^{R140Q}$-HA-NLS, pLV-TRE::NLS-IDH2$^{R140QR172K}$-HA-NLS, pLV-TRE::NLS-EGFP-T2A-NLS-mCherry) were designed using VectorBuilder online vector design tool, and synthesized by VectorBuilder Inc. To make the lentiviruses, HEK293T cells were transfected with the respective plasmids along with packaging plasmids (psPAX2 and VSVG) using jetPRIME transfection reagent (Polypus #101000015). At 48 and 72 h posttransfection, the supernatant was collected and centrifuged at $500 \times g$ for 10 min at 4 °C. The supernatant was mixed with Lenti-X™ Concentrator (Takara Bio USA, Inc.) at a 3:1 ratio and incubated overnight at 4 °C. The concentrated virus pool was centrifuged at $1500 \times g$ for 45 min at 4 °C and stored at −80 °C until further use.

### Induction of nuclear translocation of TCA cycle enzymes
iPSC-derived cardiomyocytes were transduced with the respective lentiviruses, and the medium was replaced with fresh medium after 6 h of virus addition. The cells were treated with 1 µg/mL doxycycline 48 h post-transduction to induce the expression of nuclear-targeted TCA cycle enzymes. Doxycycline was replenished every 24 h to maintain the expression. After 24 h of induction, cells were treated with 1 µM doxorubicin for 24 h. After 24 h of doxorubicin treatment, cells were harvested for the respective experiments.

### Confocal microscopy
iPSC-derived cardiomyocytes were treated with 1 µM doxorubicin for 6 or for 24 h in RPMI+ (B27+ins) medium. Post-treatment, the cells were fixed in 4% PFA for 10 min. After fixation, the cells were permeabilized with 0.25% Triton X-100 for 20 min, followed by blocking in blocking buffer (10% donkey serum, 2% BSA in 1x PBS) for 30 min. The cells were washed with 1x PBS three times and incubated with anti-PDH-E1 (1:200) (Cell Signaling 2784 S), anti-MDH2 (1:200) (Cell Signaling 11908 S), anti-IDH-2 (1:200) (Proteintech, 130-125-992), and anti-CS (1:200) (Proteintech, 16131-1-AP), Anti-γH2AX (1:500) (Cell Signaling, 9719 S), Anti-flag tag (1:200) (SIGMA, F7425), Anti-V5 tag (1:200) (Abcam, ab9137), Anti-HA tag (1:400) (BioLegend, 682404) in buffer A (2% BSA in 1x PBS with 0.05% Tween-20) overnight. The cells were washed with wash buffer A (1x PBS with 0.05% Tween-20) three times. The secondary antibodies [Donkey anti-goat Alexa-633 (1:500) (Life Technologies A21082), Donkey anti-rabbit Alexa 594 (1:500) (Life

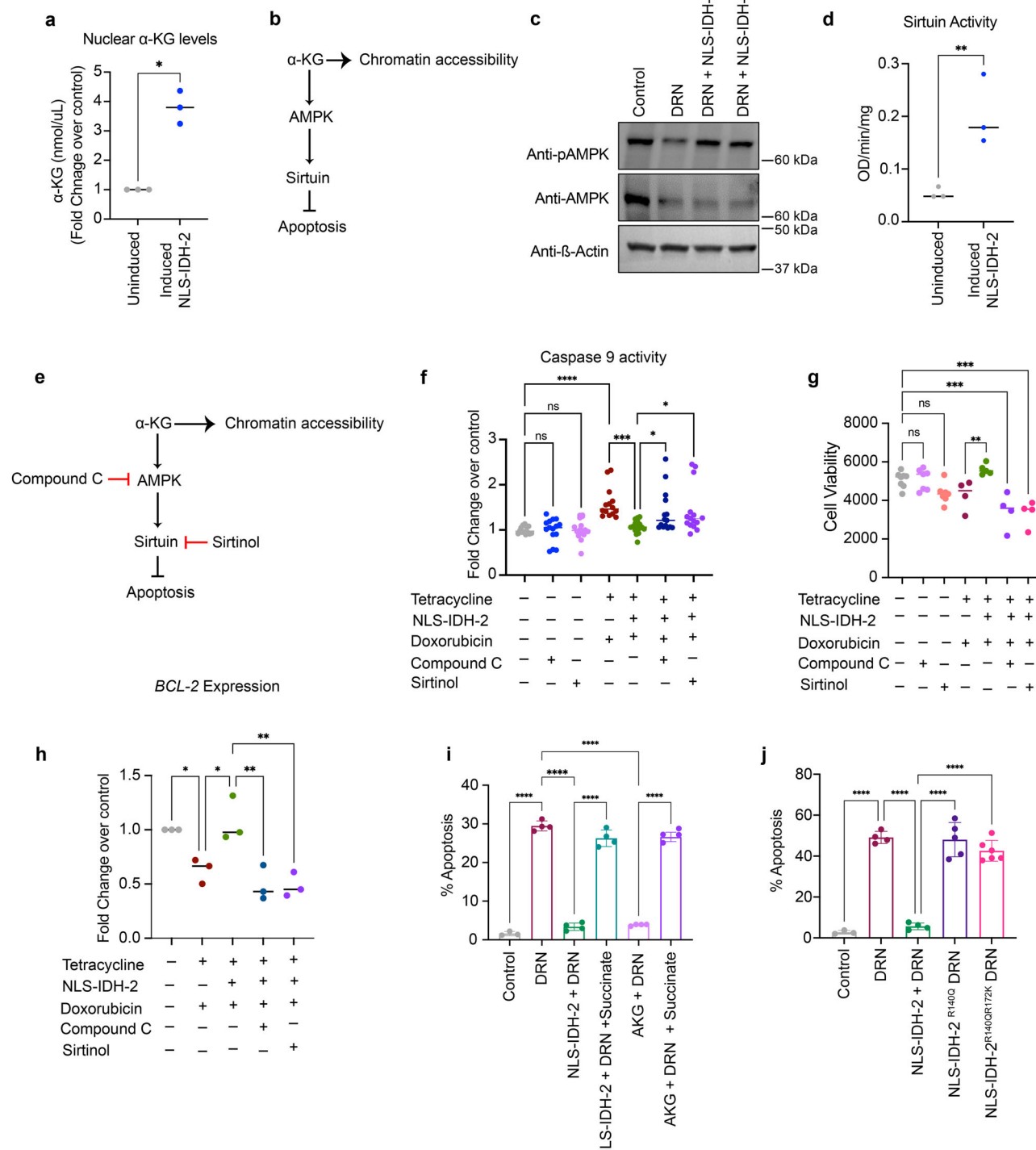

**Fig. 4 | Nuclear IDH-2 induces an anti-apoptotic effect. a** α-KG levels measured in isolated nuclei, $n = 3$ $p$ value* <0.0130. **b** Schematic of the proposed α-KG-AMPK-Sirtuin- pathway. **c** Western blots showing increased AMPK phosphorylation upon nuclear IDH-2 expression. ß-actin was used as a loading control. **d** Sirtuin activity represented as OD/min/mg. $n = 3$ $P$ value** = 0.0036. **e** Schematic of inhibitory sites of compound C and sirtinol in the α-KG-AMPK-Sirtuin pathway. **f, g** Caspase 9 activity (**f**) $n = 3$ $p$ value* <0.05, $p$ value*** = 0.0002, $p$ value **** <0.0001, and cell death (**g**) $n = 4$ $p$ value*** <0.0005, $p$ value** = 0.0085 upon inhibition of AMPK and sirtuin by compound C and sirtinol, respectively. **h** Reduced gene expression of anti-apoptotic protein *BCL2* by AMPK and sirtuin inhibition in cells expressing nuclear IDH-2. $n = 3$ $p$ value** <0.05, $p$ value* <0.002. **i, j** Annexin V staining in iPSC-CM expressing nuclear IDH-2 and treated with succinate in the presence of DRN (**i**) or expressing IDH-2 mutants in the nuclei of cells treated with DRN (**j**). $n = 3$ $P$ value**** <0.0001.

Technologies A21207), Donkey anti-mouse Alexa-488 (1:500) (Invitrogen A21202) were diluted in buffer A and incubated for 45 min at room temperature. The cells were washed with wash buffer A and then mounted on the antifade reagent ProLong Diamond with DAPI (Invitrogen™ P36962).

To detect the nuclear TCA cycle enzymes targeted into the nuclei by tetracycline-inducible lentiviruses, we fixed the cells 48 h post-tetracycline induction in 4% PFA. The cells were incubated with 95% cold methanol for 7 min and washed with 1x PBS three times. Samples were blocked in 2% BSA. Cells were then

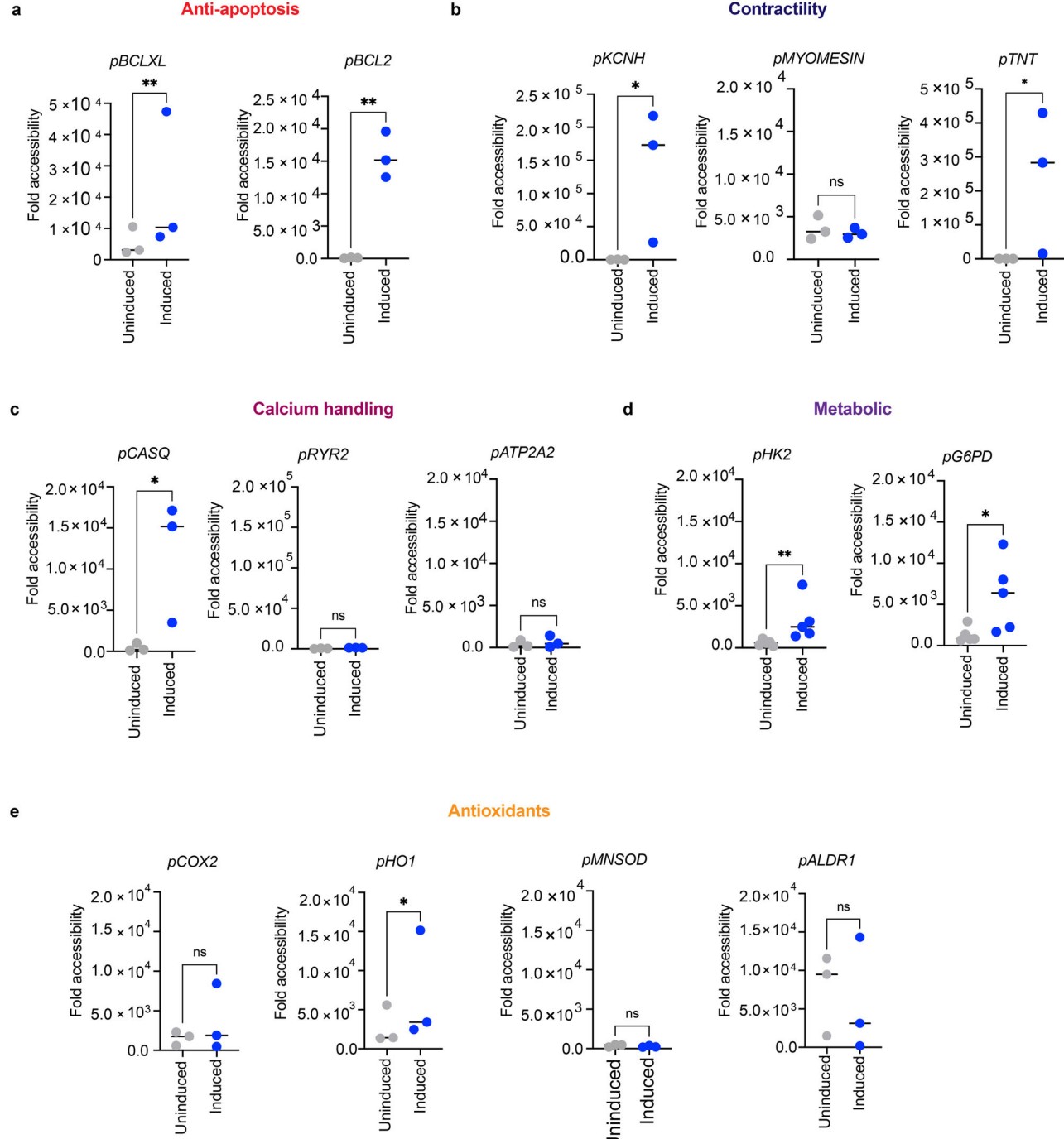

**Fig. 5 | Nuclear IDH-2 regulates chromatin accessibility. a–e** Accessibility of promoter region of cardioprotective genes ranging from anti-apoptotic (**a**), antioxidants (**b**), contractility related (**c**), involved in calcium handling (**d**), and metabolic genes (**e**) as assessed by ATAC-qPCR. $n = 3$ $P$ value** <0.01, $P$ value* <0.05.

incubated with anti-HA antibody conjugated with Alexa 647 fluorophore (1:400) in buffer B (2% BSA in 0.1% Triton X-100 in 1x PBS) for 2 h. The nonspecific antibody was washed with wash buffer B (0.1% Tween-20 in 1x PBS) three times. The samples were blocked again in 2% BSA followed by overnight incubation in anti-flag tag (1:200) and anti-V5 tag (1:200) diluted in buffer A. Post-primary antibody incubation, the cells were washed with wash buffer B and mounted on the antifade reagent ProLong Diamond with DAPI (Invitrogen™ P36962).

Images were acquired using a Zeiss LSM 710 META confocal microscope and ZEN software. The images were analyzed using Fiji-ImageJ (version: Fiji for mac OS X). The quantification of co-

localization was done using the JACoP plugin to measure the co-localization signal of mitochondrial dehydrogenase and citrate synthase with CoxIV or DAPI (staining nuclei), and the quantification is presented as Mander's co-efficient of co-localization. The 3-dimensional reconstruction was performed with an IMARIS image analysis workstation.

**Subcellular fractionation**
To isolate nuclear fractions, cells were washed with 1x PBS and scraped off from the surface, followed by centrifugation at $300 \times g$ for 3-4 min at 4 °C. The cell pellet was resuspended in 2 mL of hypotonic buffer (10 mM Tris, 10 mM NaCl, 3 mM MgCl2, 1 mM EDTA) and incubated on

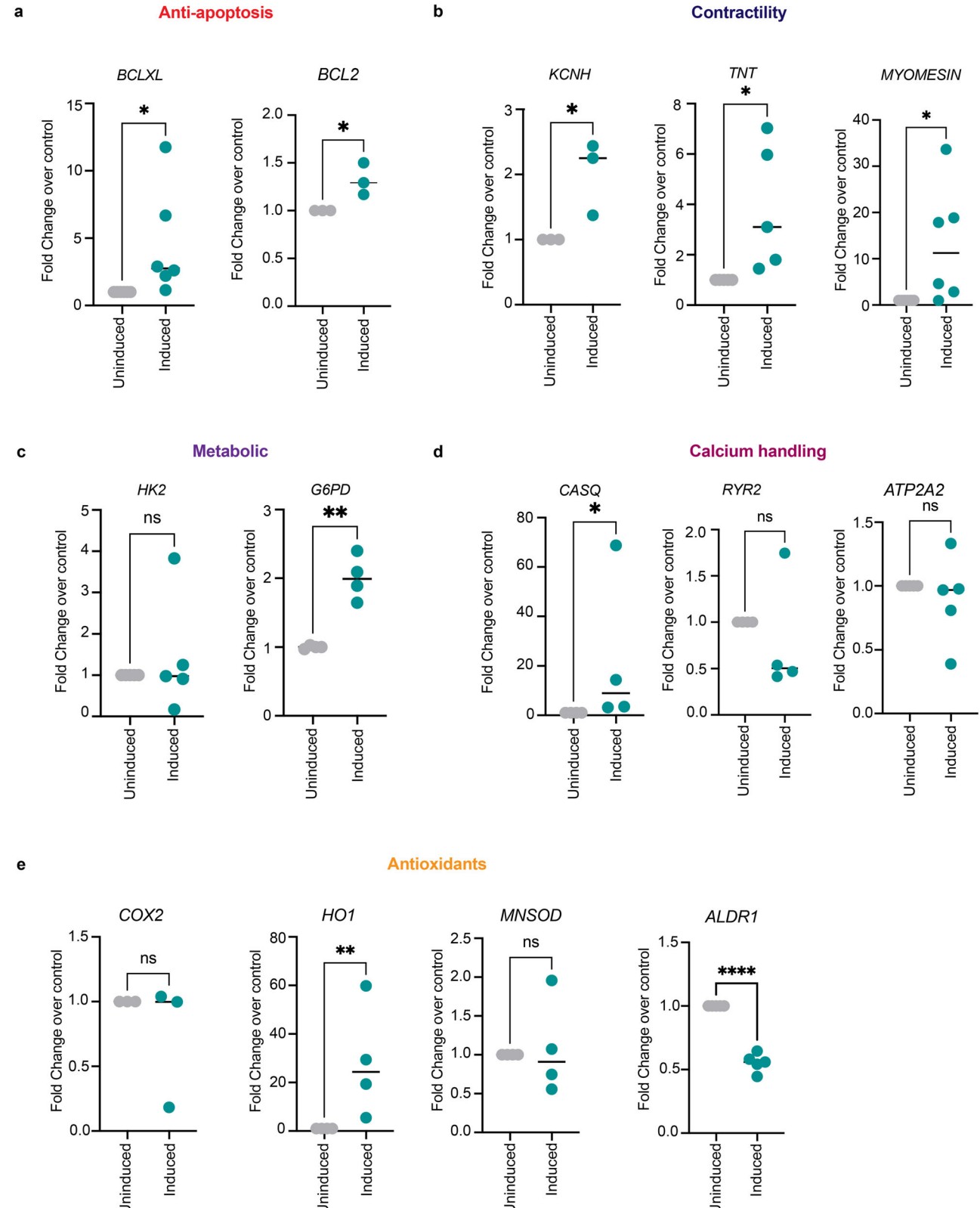

**Fig. 6 | Nuclear IDH-2 mediated changes in chromatin accessibility correspond to gene expression changes. a–e** mRNA levels of anti-apoptotic genes (**a**), contractility-related genes (**b**), metabolic genes (**c**), calcium handling (**d**), and antioxidant genes (**e**). $n = 3$ $P$ value**** <0.0001, $P$ value** <0.009, $P$ value* <0.05.

ice for 15 min. The swelled cells were then homogenized using a dounce homogenizer followed by centrifugation at $600 \times g$ for 10 min. To remove the possible contaminants, the pellet containing the nuclear fraction was resuspended in 1x PBS and centrifuged at $400 \times g$ for 2 min. The nuclear pellet was again resuspended in 1x PBS and centrifuged for 2 min at $300 \times g$. The post-spin nuclear fraction was processed either for western blotting or for enzyme assay as described in the respective sections.

## Sucrose-gradient centrifugation

Isolation of nuclei using sucrose-gradient centrifugation was performed following the published protocol[51]. Briefly, cells were harvested and washed with PBS to remove the culture medium. Cells were resuspended in hypotonic buffer and homogenized using a dounce homogenizer followed by centrifugation at $600 \times g$ for 10 min. Nuclear pellet was then resuspended in 9 volume of hypertonic sucrose buffer (2.2 M sucrose, 1 mM $MgCl_2$, 10 mM Tris-HCl, pH 7.4) and homogenized again using Dounce homogenizer on ice. The homogenized samples were centrifuged at $60,000–80,000 \times g$ for 80 min at 4 °C, the supernatant containing sucrose was removed, and the isolated nuclei were stored for further use.

## Dehydrogenase activity assays

The enzymatic activity of nuclear dehydrogenases was measured using an Isocitrate Dehydrogenase Assay Kit (Abcam, ab102528) for IDH-2, a Malate Dehydrogenase 2 (MDH2) Activity Assay (Abcam, ab119693) for MDH-2, and a Pyruvate dehydrogenase (PDH) Enzyme Activity Assay Kit (Abcam, ab109902). The nuclear fractions isolated from cells expressing nuclear IDH-2, MDH-2, or PDHC were processed according to the manufacturer's protocol to measure the enzyme activity of IDH-2, MDH-2, and PDHC.

## Western blots

Cells were lysed in RIPA buffer containing protease and phosphatase inhibitor cocktails for 30 min and centrifuged at 14k rpm for 10 min at 4 °C. For nuclear lysates, isolated nuclear fractions were lysed in nuclear lysis buffer (50 mM Tris-HCl, pH 7.4, 150 mM NaCl, 1 mM EDTA, 0.5% Triton X-100, 1 mM DTT, and protease and phosphatase inhibitor cocktails). The samples for SDS–PAGE were prepared in 4x Laemmli buffer containing ß-mercaptoethanol and boiled at 92 °C for 10 min. Following separation by SDS–PAGE, the proteins were transferred to PVDF membranes and immunoblotted with the respective primary and secondary antibodies. The primary antibodies used were anti-IDH-2 (1:1000) (Proteintech, 23254-1-AP), anti-PDH-E1 (1:1000) (Cell Signaling 2784 S), anti-MDH-2 (1:1000) (Cell Signaling 11908 S), anti-PCNA (1:1000) (Cell Signaling 13110), anti-phospho AMPKα (1:1000), anti-AMPKα (1:1000) (Cell Signaling 2532 S), anti-ßactin (1:5000) (Santa Cruz SC-47778 HRP), anti-caspase 9 (1:500) (NOVUS Biologicals NB100-56118), anti-BCL-xL (1:1000) (NOVUS Biologicals NB100-56104), and anti-MCl1 (1:1000) (NOVUS Biologicals NB100-56146), anti-Histone H3 (1:1000) (Cell Signaling 96C10), H3K79me2 (1:1000) (Invitrogen 710802), H3K4me3 (1:1000) (Invitrogen MA5-11199, monoclonal antibody clone G.532.8), and H3K36me3 (1:1000) (Invitrogen PA5-17109), Anti-cardiac troponin T (1:1000) (Thermo Fisher, MA5-12960, monoclonal antibody clone 13-11), Anti-kir2.1 (1:1000) (R&D Systems, MAB9548, monoclonal antibody clone # 2153 C), Anti-flag tag (1:1000) (SIGMA, F7425), Anti-V5 tag (1:1000) (Abcam, ab9137), Anti-HA tag:HRP (1:1000) (Cell Signaling 6E2 monoclonal antibody clone #2999). The secondary antibodies used in this study are Peroxidase AffiniPure F(ab')₂ Fragment Goat Anti-Rabbit IgG (H + L) (Jackson ImmunoResearch Laboratories Inc. 111-036-003) and Peroxidase AffiniPure Goat Anti-Mouse IgG, light chain specific (Jackson ImmunoResearch Laboratories Inc. 115-035-174).

## Annexin V staining

The iPSC-derived cardiomyocytes were subjected to doxorubicin treatment or lentiviral transduction and doxorubicin treatment as per the experimental requirements. In the experiments where reduction in doxorubicin-cardiotoxicity by nuclear TCA cycle enzymes was assessed, doxorubicin-treated cells experienced equal amounts of tetracycline as cells with induced nuclear targeting of TCA cycle enzymes. Cells were harvested and stained according to the manufacturer's protocol (Bio-Rad), followed by analysis of cells positive for annexin V-FITC by using a BD LSRFortessa™ Flow Cytometer. Data were analyzed using BD FACSDiva™ software (version 6.2). The gating strategy is shown in supplementary fig. 13.

## qRT–PCR

To isolate RNA, cells were lysed in QIAzol Lysis Reagent, and RNA was isolated using the miRNeasy Mini Kit (Qiagen 217004). cDNA was synthesized from RNA in a reverse transcription reaction using a High Capacity cDNA Reverse Transcription Kit (Thermo Scientific 4368814). qRT–PCR was performed using FastStart Universal SYBR Green Master Mix (Rox) (Roche 4913850001) on a QuantStudio 7 Flex real-time PCR detector (Thermo Fisher) with the primers. About 10 ng of RNA and 10 μM of the primer were used for each reaction. Quantification of relative gene expression was normalized to B2M and GAPDH.

## Troponin I assay

iPSC-derived cardiomyocytes were treated with doxorubicin or transduced with the NLS constructs prior to doxorubicin treatment. After 24 h of doxorubicin treatment, media was collected from each well and filtered with a 0.22 μm filter to remove cellular debris. The released troponin I was measured using a Troponin I (Human) ELISA Kit (Abnova, KA0233) according to the manufacturer's protocol.

## Cardiomyocyte contractility assay

hiPSC-CMs were transduced with NLS-IDH2, and contractility was analyzed 24 h after DRN treatment in comparison with cells not treated with DRN. Beating patterns were captured at 25 frames per second (fps) over 20 s for each position on a Zeiss TIRF microscope with a bright field using a 10X objective. At least 30 positions were captured while maintaining the temperature at 37 °C and $CO_2$ at 5%. The contraction velocity and beating rates were analyzed using the MATLAB-based Conklin motion tracking algorithm[52]. Briefly, images were divided into arrays of pixel macroblocks, and the motion of each macroblock for subsequent frames was calculated through a single-sweep exhaustive search block matching algorithm. By applying the vector amplitude, motion velocity tracings are generated, featuring typical contraction and relaxation peaks and allowing for the calculation of parameters such as beat rate and peak velocities.

## Calcium transients

For calcium imaging, hiPSC-CMs were reseeded in Matrigel-coated glass-bottom dishes, and calcium transient measurements were assessed using the Fluo-4 Direct Calcium Assay kit (Invitrogen, F10471) according to the manufacturer's instructions. Briefly, 2x Fluo-4 Direct calcium reagent loading solution was added to wells containing cells in culture medium and incubated at 37 °C for 10 min. Then, the media was removed and replaced with fresh growth media. Calcium imaging was conducted using a Zeiss LSM 710 BIG confocal microscope with a 63x objective. Subsequent line scans were recorded with excitation at 488 nm.

## Transthoracic echocardiography

Transthoracic echocardiography was performed on unconscious mice following DRN or PBS treatment to determine systolic left ventricular function. Anesthesia was induced using 3% isoflurane supplied to an induction chamber, whereafter, the lack of hind paw response was used to confirm induction. Mice were then secured to a warming table to maintain body temperature, where hair removal was done with the use of depilatory cream. Concentrations of isoflurane were maintained between 1–1.5% to maintain a target heart rate of $450 \pm 50$ BPM. Ultrasound scans were obtained with the Vevo2100 imaging system (version 2.2.0(Build 12089)) (Visual Sonics Inc, Toronto, ON, Canada) using the MS550D probe with a center frequency of 40 MHz. M-mode

tracings were measured from the short-axis view at the midventricular level, as indicated by the presence of the papillary muscle, to measure left ventricular function and dimensions. Data were analyzed using Vivo Lab 3.2.0 (Visual Sonics Inc, Toronto, ON, Canada). Upon completion of echocardiographic scans, all mice were carefully monitored to ensure recovery from anesthesia.

### ATAC-qPCR
The chromatin accessibility of selected promoter regions was analyzed using the EpiQuik Chromatin Accessibility Assay Kit (Epigentek, P-1047-48) according to the manufacturer's instructions using iPSC-CMs transduced with NLS-IDH2 with or without doxycycline induction.

### Luminescence-based cell death assay
Cell death in response to DRN treatment or nuclear expression of TCA cycle dehydrogenases was measured using Cell Titer Glo™ (Promega, G7570) according to the manufacturer's instructions.

### Caspase 9 activity assay
To measure Caspase 9 activity in the presence of compound C and sirtinol, we used the Caspase9Glo™ assay (Promega, G8210) and followed the protocol suggested by the manufacturers.

### Metabolite measurements
Nuclei were isolated from cells with nuclear expression of IDH-2 and PDHC or cells treated with DRN as per the experimental requirements. As per the requirement of the protocol, the samples were prepared using assay buffer or were deproteinated using 13% TCA and precipitated at 15,000×$g$ for 5 min. The pH of the supernatant was adjusted with 2 N KOH to 7.5–8. The concentration of alpha-ketoglutarate was measured using an Alpha Ketoglutarate (alpha KG) Assay Kit (Abcam, ab83431). Acetyl-CoA was measured using a PicoProbe™Acetyl-CoA Fluorometric Assay Kit (BioVision, K317). Pyruvate, malate, succinate, and fumarate were measured using a Pyruvate Assay Kit (Sigma, MAK332-1KT), Malate Colorimetric Assay Kit (BioVision, K637), Succinate Assay Kit (Colorimetric) (Abcam, ab204718), and Fumarate Colorimetric Assay Kit (BioVision, K633), respectively.

### DNA methylation
Genomic DNA was isolated from cells with nuclear expression of IDH-2, and DNA methylation was measured by a MethylFlash Global DNA Methylation (5-mC) ELISA Easy Kit (P-1030-48) according to the manufacturer's guidelines.

### Histone methylation
The protein precipitated from the samples used to measure α-KG was used to analyze the histone methylation status through western blot to better understand the correlation of nuclear α-KG and its effect on histone methylation. The TCA-precipitated protein was washed with acetone and dissolved in 4x sample loading dye. The samples were boiled at 90 °C for 10 min. The blots were probed for histone H3 (Cell Signaling 96C10), H3K79me2 (Invitrogen 710802), H3K4me3 (Invitrogen MA5-11199), and H3K36me3 (Invitrogen PA5-17109).

### In vivo doxorubicin exposure and gene delivery
DRN treatment: C57/BL6 mice were injected with two doses of 10 mg/kg DRN (i.p.) (total 20 mg/kg) separated by 48 h. Mice were sacrificed 24 h after the second dose of DRN (72 h total time) for detection of TCA cycle enzymes in the nuclei of cardiomyocytes. ECHO was performed 46 h post-DRN treatment (92 h total time) to measure the effect of DRN on cardiac function. PBS-treated mice were used as a control.

NLS-IDH2 expression: Lentiviruses containing α-MHC::NLS-IDH-2-HA or α-MHC::EGFP were injected through retro-orbital intravenous injection in two doses separated by 72 h between each dose before DRN treatment. DRN was administered through intraperitoneal injection 48 h post second dose of lentiviral injection. The doxorubicin dose was the same as mentioned above. Mice were subjected to echocardiography and sacrificed to collect tissue samples 48 h after the last dose of DRN (a total of 92 h of DRN treatment).

### Statistical analysis and software
Experiments were performed with at least $n = 3$ for each human iPSC line, and the statistical tests were performed using GraphPad Prism software (version 9.1.1(223)). One-way ANOVA was used when more than two samples were compared in a data set. To assess the statistical significance of the two samples, a two-tailed Student's $t$-test was performed. To test the significance of nuclear translocation, two-way ANOVA was used.

### Reporting summary
Further information on research design is available in the Nature Portfolio Reporting Summary linked to this article.

## Data availability
All data needed to reproduce the results are available within the manuscript supplementary information and source data provided with this paper. Other relevant data is available from the corresponding authors. Source data are provided with this paper.

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

## Acknowledgements

This study was supported by NIH grants R01-HL126516 (J.R.), R01-HL152515 (J.R.), NIH R00-HL130416 and R01-HL148756 (S.-G.O.), T32 HL007829 (J.J) and American Heart Association grants 18PRE34070092 (P.G.), and 23POST1029855 (G.Y.). We would like to thank Mr. Jiajun Li for their technical assistance.

## Author contributions

S.S., S.-G.O., and J.R. conceived the idea and designed the experiments. S.S., P.G., H.M., G.Y., Y.K., J.J., P.T.T., and A.J. performed the experiments and analyzed the data. S.S., P.G., J.J., P.T.T., S.-G.O., and J.R. interpreted the data. All authors provided input during the writing or revision of the manuscript.

## Competing interests

The authors declare no competing interests.
