## [Peer Review File · Nature Communications]

Nuclear translocation of mitochondrial dehydrogenases as an adaptive cardioprotective mechanismEditorial Note: This manuscript has been previously reviewed at another journal that is not operating a transparent peer review scheme. This document only contains reviewer comments and rebuttal letters for versions considered at Nature Communications.

REVIEWERS' COMMENTS

Reviewer #1 (Remarks to the Author):

The authors have made notable improvements by providing additional evidence for the nuclear localization of the dehydrogenases and expanding the range of ATAC-qPCR targets to demonstrate the impact of nuclear IDH-2 on chromatin accessibility. Please list the primers used for ATAC-qPCR in a supplementary table.

Reviewer #2 (Remarks to the Author):

In this manuscript, the authors reported adaptive mechanisms of cardiomyocytes in response to doxorubicin induced cardiotoxicity. They observed translocation of select mitochondrial TCA cycle proteins including PDH-E1, MDH-2, and IDH-2 both in vitro using iPSC-derived cardiomyocytes and in vivo. They then showed lentivirus induced nuclear expression of these mitochondrial enzymes prevented doxorubicin-induced cardiomyocyte cell death as well as DNA injury.

Overall, I believe the authors satisfactorily addressed my previous comments, as described below:

1. I previously requested additional experiment regarding what would happen to mice injected with NLS-IDH2 vs control with PBS to see whether NLS-IDH2 would improve LVEF baseline in the absence of doxorubicin. The authors appropriately addressed this concern and showed that there is no functional change at baseline between NLS-IDH2 vs PBS based on LVEF and FS changes.

2. In my request to address rationale behind picking select genes (KCNH, CasQ, TnT) for

ATAC-qPCR, the authors first categorized genes based on their functional consequences (e.g. anti-apoptosis, anti-oxidant, contractility, calcium handling, and metabolism) and showed that select genes in these categories with newly open chromatin resulted in increased RNA expression at 24 hour mark, as presented in new Figure 5 (although as the authors pointed out, some genes with open chromatin didn't necessarily result in increased RNA expression).

I assume it might be due to the effort to fit all figures into one, the current Figure 5 is a bit difficult to follow. I suggest to show ATAC then mRNA qPCR results for the each category.

3. In our previous comment, we raised concern regarding 1) unusual/inconsistent ventricular beating rates of iPSC-CMs and 2) contraction velocity that appears to suggest improved, not worsened contraction, after IDH2 trans-location.

In response to this, the authors first commented that baseline iPSC-CM beating rates can be as high as 100s. While I do wonder if the baseline beating rate is in the range of 100s, it may resemble more atrial-like CM, I see that the contraction figure on 3C, the beating rate is only 48 BPM (16 peaks in the span of 20 seconds).

Additionally, in addition to my comment on contraction/relaxation velocity, the authors also included deformation distance which showed significant decrease in response to DRN with near normalization with IDH-2.

Response to the reviewers

Reviewer 1

The authors have made notable improvements by providing additional evidence for the nuclear localization of the dehydrogenases and expanding the range of ATAC-qPCR targets to demonstrate the impact of nuclear IDH-2 on chromatin accessibility. Please list the primers used for ATAC-qPCR in a supplementary table.

Response: We thank the reviewer for their suggestions and we have included the requested primer table in the revised manuscript.

Reviewer 2:

In this manuscript, the authors reported adaptive mechanisms of cardiomyocytes in response to doxorubicin induced cardiotoxicity. They observed translocation of select mitochondrial TCA cycle proteins including PDH-E1, MDH-2, and IDH-2 both in vitro using iPSC-derived cardiomyocytes and in vivo. They then showed lentivirus induced nuclear expression of these mitochondrial enzymes prevented doxorubicin-induced cardiomyocyte cell death as well as DNA injury.

Overall, I believe the authors satisfactorily addressed my previous comments, as described below:

1. I previously requested additional experiment regarding what would happen to mice injected with NLS-IDH2 vs control with PBS to see whether NLS-IDH2 would improve LVEF baseline in the absence of doxorubicin. The authors appropriately addressed this concern and showed that there is no functional change at baseline between NLS-IDH2 vs PBS based on LVEF and FS changes.

Response: We thank the reviewer for accepting our revisions.

2. In my request to address rationale behind picking select genes (KCNH, CasQ, TnT) for ATAC-qPCR, the authors first categorized genes based on their functional consequences (e.g. anti-apoptosis, anti-oxidant, contractility, calcium handling, and metabolism) and showed that select genes in these categories with newly open chromatin resulted in increased RNA expression at 24 hour mark, as presented in new Figure 5 (although as the authors pointed out, some genes with open chromatin didn't necessarily result in increased RNA expression).

I assume it might be due to the effort to fit all figures into one, the current Figure 5 is a bit difficult to follow. I suggest to show ATAC then mRNA qPCR results for the each category.

Response: We have now split **Figure 5** into two figures by separating ATAC-qPCR (**new Fig 5**) and mRNA-qPCR (**new Fig 6**).

3. In our previous comment, we raised concern regarding 1) unusual/inconsistent ventricular beating rates of iPSC-CMs and 2) contraction velocity that appears to suggest improved, not worsened contraction, after IDH2 trans-location.

In response to this, the authors first commented that baseline iPSC-CM beating rates can be as high as 100s. While I do wonder if the baseline beating rate is in the range of 100s, it may

resemble more atrial-like CM, I see that the contraction figure on 3C, the beating rate is only 48 BPM (16 peaks in the span of 20 seconds).

Additionally, in addition to my comment on contraction/relaxation velocity, the authors also included deformation distance which showed significant decrease in response to DRN with near normalization with IDH-2.

Response: We thank the reviewer for accepting our revisions.